# Towards Automatic Evaluation and Selection of PHI De-identification Models via Multi-Agent Collaboration

**AI Scientist, Guanchen Wu[1], Zuhui Chen[2], Yuzhang Xie[1], Carl Yang[1]**
[1]Department of Computer Science, Emory University
[2]Department of Statistics, Columbia University
{guanchen.wu, yuzhang.xie, j.carlyang}@emory.edu
zc2748@columbia.edu

## Abstract

Protected health information (PHI) de-identification is critical for enabling the safe reuse of clinical notes, yet evaluating and comparing PHI de-identification models typically depends on costly, small-scale expert annotations. We present TEAM-PHI, a *multi-agent evaluation and selection framework* that uses large language models (LLMs) to automatically measure de-identification quality and select the best-performing model without heavy reliance on gold labels. TEAM-PHI deploys multiple *Evaluation Agents*, each independently judging the correctness of PHI extractions and outputting structured metrics. Their results are then consolidated through an LLM-based majority voting mechanism that integrates diverse evaluator perspectives into a single, stable, and reproducible ranking. Experiments on a real-world clinical note corpus demonstrate that TEAM-PHI produces consistent and accurate rankings: despite variation across individual evaluators, LLM-based voting reliably converges on the same top-performing systems. Further comparison with ground-truth annotations and human evaluation confirms that the framework's automated rankings closely match supervised evaluation. By combining independent evaluation agents with LLM majority voting, TEAM-PHI offers a practical, secure, and cost-effective solution for *automatic evaluation and best-model selection* in PHI de-identification, even when ground-truth labels are limited. [1]

## 1 Introduction

Clinical notes—such as discharge summaries, nursing notes, and radiology or ECG reports—capture rich context like clinical reasoning, observations, and social factors that are often missing in structured electronic health records [Seinen et al., 2025, Tayefi et al., 2021, Zhang et al., 2024]. Despite their value for research and downstream applications, clinical notes often contain protected health information (PHI), including names, dates, and addresses, and their reuse is strictly regulated under privacy laws such as HIPAA [Moore and Frye, 2019, Cohen and Mello, 2018]. To enable safe sharing and analysis, PHI de-identification is typically applied as a preprocessing step, automatically detecting and removing or replacing personal identifiers—for example, transforming "John Smith was admitted on 03/15/2024 at Hospital A" into "[NAME] was admitted on [DATE] at [HOSPITAL]".

Existing PHI de-identification (De-id) systems have progressed from handcrafted rules and feature-based machine learning to deep learning and transformer-based models [Kovačević et al., 2024]. More recently, large language models (LLMs) have opened new avenues for clinical NLP tasks such

---

[1]The implementation details, prompt designs, and codes are available in the anonymous repository at https://anonymous.4open.science/r/SAFE-0915.

as named entity recognition, relation extraction, and note summarization [Lehman et al., 2023, Wu et al., 2024]. Studies show that multiple kinds of LLMs can effectively extract PHI entities, generate structured representations from free text, and assist in clinical decision-making [Wu et al., 2025, Liu et al., 2023]

Although LLM-based PHI de-identification has significantly advanced the field, methods for systematically evaluating and automatically selecting the best de-identification model remain underexplored. Conventional evaluation relies on costly, small, institution-specific expert annotations, which impede the creation of reliable, generalizable benchmarks [Altalla' et al., 2025]. LLMs have shown broad prior knowledge and strong zero-shot capabilities [Chang et al., 2024], offering a promising route to assess de-identification models without gold labels. However, using LLMs as verifiers raises new challenges: how to design efficient, calibrated, and robust evaluation pipelines that are insensitive to prompt phrasing or choice of verifier, and how to ensure their judgments are trustworthy across use cases [Chang et al., 2024, de Oliveira et al., 2025].

In this paper, we propose TEAM-PHI (Trusted Evaluation and Automatic Model selection for PHI), a multi-agent framework to automatically evaluate LLM-based de-identification methods and select the best-performing model. TEAM-PHI first runs multiple LLMs as De-id models that extract PHI from raw clinical text in a structured format. Multiple evaluation agents then assess those De-id outputs without relying on gold annotations. We validate the framework on a real-world clinical-note corpus and find that (1) aggregating Evaluation Agents by majority voting improves reliability relative to single-model assessments, (2) the multi-agent evaluators operate efficiently and produce consistent relative rankings, and (3) Llama-70B consistently emerges as the most reliable De-id model — a result supported by ablation studies using a limited manually annotated test set and independent human expert review.

Our contributions are threefold: (i) we introduce TEAM-PHI, a multi-agent framework that uses multiple LLMs to perform automated PHI evaluation and select the best de-identification model without heavy reliance on human labels; (ii) we demonstrate through extensive experiments that the multi-agent design yields consistent and accurate estimates of de-identification quality and that voting aggregation improves evaluator reliability; and (iii) we show that TEAM-PHI is a practical, secure, and cost-effective tool for guiding deployment of privacy-preserving data pipelines in healthcare settings.

## 2 Related Work

**PHI De-identification.** De-identification of PHI has evolved from rule-based systems with hand-crafted dictionaries and regexes [Uzuner et al., 2007, Meystre et al., 2010] to statistical learners (CRFs, SVMs) [He et al., 2015, Jiang et al., 2017], then to deep models (BiLSTM-CRF) [Dernoncourt et al., 2017, Tang et al., 2020] and transformer-based architectures (e.g., BioBERT) [Lee et al., 2020, Johnson et al., 2020], improving accuracy while reducing manual feature engineering. Recently, LLMs have been leveraged for many clinical tasks [Xie et al., 2024, Wu et al., 2024], including PHI de-identification, because of their strong contextual reasoning and zero-/few-shot learning abilities [Bhasuran et al., 2025]. Recent work has demonstrated the promise of large language models (LLMs) for PHI De-id. For example, DeID-GPT leverages GPT-4 for zero-shot de-identification of medical text, achieving competitive performance without task-specific fine-tuning [Liu et al., 2023]. Similarly, Wu et al. [2025] proposed the LPPA framework, which combines synthetic data generation with instruction tuning to fine-tune local LLMs for PHI extraction. LPPA offers a privacy-preserving solution while notably improving recall—a critical metric for safeguarding sensitive health data—and its hybrid models (LPPA-4K and LPPA-5K) are also adopted in our implementation of De-id models. Despite these advances, existing systems still lack robust methods for systematically evaluating and automatically selecting the best De-id model, motivating the development of our proposed framework.

**Evaluation of De-identification Systems.** Traditionally, PHI de-identification systems are evaluated against gold-standard corpora such as the i2b2 shared tasks [Stubbs et al., 2015]. Metrics typically include precision, recall, and F1-score. However, such datasets are costly to create and limited in scope. Alternative evaluation approaches include weak supervision, ensemble frameworks, and extensible benchmarking systems designed to assess de-identification tools across multiple corpora and settings [Heider and Meystre, 2024]. Despite recent progress, robust evaluation without gold-standard annotations, as in this study, is still largely unexplored.

**Large Language Models as Judges.** Some researchers has begun to examine the use of LLMs as evaluators, or "judges," in a range of NLP tasks including summarization, dialogue, and machine translation. These studies show that LLMs can approximate human judgments with promising consistency [Zhu et al., 2023, Hada et al., 2023, Xie et al., 2025, Pan et al., 2024]. Specialized evaluator models such as Prometheus incorporate rubric-based scoring to align more closely with human ratings [Kim et al., 2023]. However, concerns of bias, calibration, and fairness remain when deploying LLM-based judges at scale [Wang et al., 2023]. Our work extends this line of research to the sensitive domain of PHI de-identification, systematically evaluating whether Evaluation Agents can replicate gold-standard and human assessments when ground truth is limited or masked.

## 3 Method

### 3.1 Problem Formulation

**Notation.** All notations used throughout the paper are summarized in Appendix A.

**Task Definition.** Let $\mathcal{X}$ denote the space of clinical notes, and let $\mathcal{C} = \{c_1, \ldots, c_m\}$ be the finite set of PHI categories (e.g., PERSON, DATE/TIME, LOCATION, ORGANIZATION). For a given note $x \in \mathcal{X}$, the de-identification output is a set of labeled entity spans:

$$\text{PHI}(x) = \{(c_i, e_i)\}_{i=1}^k, \tag{1}$$

where $c_i \in \mathcal{C}$ and $e_i \subseteq x$ is a contiguous text span corresponding to an entity of type $c_i$. Our objective is to select the best-performing de-identification model from a candidate set $\mathcal{M} = \{M_1, \ldots, M_R\}$ by maximizing a utility function that does not require gold-standard annotations:

$$M^* = \arg\max_{M \in \mathcal{M}} \mathbb{E}_{x \sim \mathcal{D}} \text{ PHI-EV}(M, x), \tag{2}$$

where $\text{PHI-EV}(M, x)$ denotes the evaluation score for model $M$ on input $x$.

**Conventional Evaluation.** In traditional settings, given a model $M$, its prediction $\text{PHI}_M(x)$, and the corresponding gold annotations $y^*(x)$, the evaluation score is computed via a standard metric:

$$\text{PHI-EV}(M, x) = f\big(\text{PHI}(M, x), y^*(x)\big), \tag{3}$$

where $f$ is typically precision, recall, or F1-score.

**TEAM-PHI Evaluation.** Due to the scarcity of large-scale annotated corpora $\{(x, y^*(x))\}$ in real-world applications, we propose replacing $f$ with a multi-agent evaluation function $g$ that aggregates judgments from $A$ independent LLM-based evaluation agents $\{E_a\}_{a=1}^A$:

$$\text{PHI-EV}(M, x) = g\big(\text{PHI}(M, x)\big), \qquad g = \text{Aggregate}(E_1, \ldots, E_A) \quad \text{(e.g., majority voting)}. \tag{4}$$

### 3.2 Framework Overview

To meet this objective, we propose a multi-agent evaluation framework 1 that decouples PHI extraction from quality assessment. First, raw clinical notes are processed in parallel by multiple De-id models, each producing a set of PHI predictions. These predictions are then evaluated by a pool of independent *Evaluation Agents*—LLM-based judges that assess the correctness of every predicted PHI entity without relying on gold annotations. Their judgments are aggregated through majority voting and further verified by targeted human review to generate trustworthy model comparisons and rankings. This modular design supports large-scale benchmarking of de-identification systems without human-labeled corpora and can flexibly incorporate new de-identification or evaluation models without altering the overall protocol.

### 3.3 De-id Models

We employ a diverse set of De-id models—eight LLMs, including two LPPA hybrids trained on synthetic clinical notes [Wu et al., 2025]. Each model receives the same prompt and outputs predictions as a JSON dictionary of PHI entities, ensuring a unified, machine-readable format across heterogeneous architectures. Although these models differ in scale, training data, and internal reasoning, our framework treats them purely as black boxes and focuses on rigorous and reproducible evaluation and selection through multi-agent collaboration. The process are described in Eq.(3).

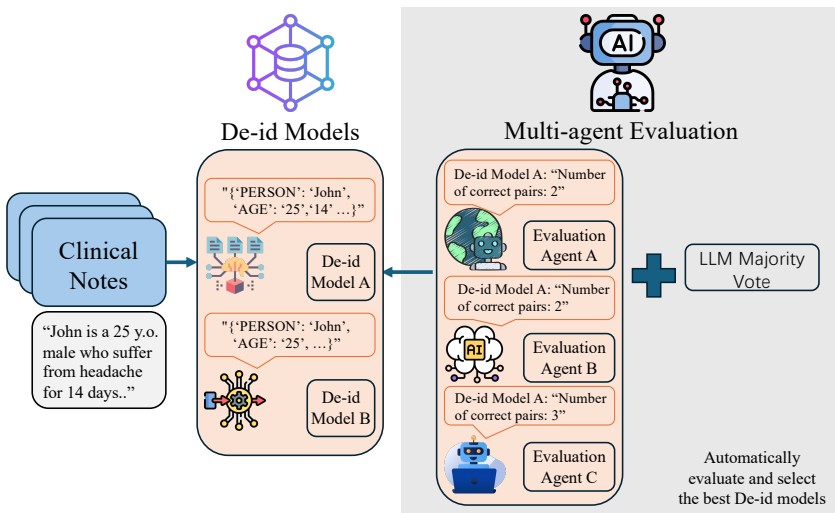

Figure 1: Framework of TEAM-PHI.

## 3.4 Multi-agent Evaluation

Evaluation Agents serve as independent judges of the quality of PHI de-identification. Given a clinical note $x$ and the predicted output $\text{PHI} = \{(c_1, e_1), \cdots, (c_i, e_i)\}$, the Evaluation Agents $\{E_a\}_{a=1}^A$ will evaluate De-id model outputs and select the best performing model follow Eq.(4). Evaluation Agents face challenges distinct from those of De-id models, and our design addresses them explicitly.

**Normalization of linguistic variation.** PHI entities may appear in multiple surface forms, such as different name prefixes, diverse date formats, or varied expressions of age. To handle these cases we introduce a normalization procedure that standardizes entity spans (e.g., mapping "Dr. Smith" and "Smith" to the same PERSON mention, or converting "03/15/2024" and "March 15, 2024" to a canonical date).

**Consistent and machine-readable output.** To ensure reproducibility and facilitate automated scoring, each Evaluation Agent is constrained to output its decision as a strict JSON object of the form `{"Number of Correct Pairs":N}`, where $N$ is the integer total of validated predictions. This removes ambiguity caused by formatting differences and allows direct aggregation of results across agents.

**Mitigating evaluator bias.** While De-id models must identify all PHI entities, Evaluation Agents must critically verify predictions without over-accepting incorrect ones or missing valid but differently expressed entities. To reduce individual bias and quantify reliability, multiple Evaluation Agents are deployed in parallel, and their outputs are combined through ensemble strategies such as majority voting.

**LLM Majority Voting.** To consolidate the independent judgments from multiple Evaluation Agents into a single, reproducible ranking of De-id models, we introduce an LLM majority voting mechanism. This mechanism treats each Evaluation Agent's summary of correctness counts as a "ballot" and uses an LLM to aggregate these ballots. Two complementary modes are employed: (i) *Independent voting* in which the LLM inspects one Evaluation Agent's summary table at a time and nominates the best-performing De-id model based on that single table; the final winner is determined by majority across all tables. (ii) *Cross-informed voting* in which the LLM simultaneously reviews all Evaluation Agents' tables and chooses a single best model after considering the combined evidence.

This dual-mode design serves several purposes. First, independent voting captures diverse evaluator perspectives and guards against a single evaluator dominating the decision. Second, cross-informed voting enables the LLM to reason globally across evaluators, potentially identifying consensus patterns or compensating for individual evaluator noise. By applying natural-language reasoning to structured numerical evidence, LLM majority voting provides an automated and explainable way to

integrate heterogeneous assessments. It does not re-evaluate the clinical content itself but instead transforms multiple correctness counts into a reliable and reproducible model ranking.

Together, these design elements enable the framework to automatically evaluate and select the best-performing De-id models, even when gold-standard annotations are masked, while maintaining transparency and reproducibility in the aggregation process.

## 4 Experimental Settings

Our experiments evaluate the proposed multi-agent framework for PHI de-identification in clinical notes. We focus on two questions: (1) can Evaluation Agents provide consistent and reliable judgments of de-identification quality when gold-standard annotations are hidden, and (2) how well do these judgments align with supervised evaluation and human assessment.

### 4.1 Dataset

We use a dataset of 100 fully annotated clinical notes provided by a large U.S. hospital. Each note averages about 1,000 tokens and contains a diverse set of PHI mentions that were meticulously annotated by medical experts. While gold-standard labels are available, they are masked in the main experiments and used only for a final supervised check (Table 8).

### 4.2 LLM Choices

Eight LLMs act as De-id models, including commercial APIs and open-source models, as well as two hybrid LPPA models trained on synthetic clinical notes [Wu et al., 2025]. Six LLMs are employed as Evaluation Agents. To ensure fair comparison, all De-id models receive the same prompt and must output a structured JSON dictionary of predicted PHI entities, and Evaluation Agents judge these outputs and return the number of correct predictions in a fixed JSON format.

For readability in our results tables, we use short model names such as *Mistral-7b*, *GPT-3.5*, *GPT-4o*, *Llama-8b*, and *Llama-70b*.

More implementation details—including dataset details, model links, evaluation metrics, validation details, and compute resources—are provided in the Appendix B.

## 5 Experimental Results

### 5.1 TEAM-PHI Evaluation Results

**Single Agent Evaluation.** Tables 1 to 6 report results when each of the six Evaluation Agents serves as the judge. Although absolute scores vary considerably across Evaluation Agents, several consistent patterns emerge. Llama-70B is repeatedly judged as one of the top-performing De-id models, appearing in the top-2 in 2 of 6 evaluations ($\approx$33%) and in the top-3 in 4 of 6 evaluations ($\approx$67%). It maintains a high average Recall-Proxy ($\approx$0.62), strong *Num Correct* values (mean$\approx$1,018), and consistently achieves $\geq$0.80 in PERSON recognition, underscoring its stable, evaluator-independent performance. GPT-4o also demonstrates strong performance, particularly in recall-proxy and DATE/TIME recognition. In contrast, smaller models such as GPT-3.5 and Gemma-2 show greater variability: GPT-3.5's Recall-Proxy spans 0.27–0.87 (mean$\approx$0.66, std$\approx$0.19), while Gemma-2 spans 0.26–0.69 (mean$\approx$0.53, std$\approx$0.13). Hybrid models tend to achieve higher precision but exhibit lower coverage, suggesting a trade-off between accuracy and completeness.

**Entity-Level Performance.** The entity-specific metrics in Tables 1 to 6 highlight additional insights. For the PERSON category, several models, including Llama-70B and GPT-4o, consistently achieve high recognition rates, often exceeding 0.90. This indicates that models are well attuned to detecting personal identifiers such as names. By contrast, performance on DATE/TIME varies more substantially. Some Evaluation Agents judge GPT-4o and Llama-8B as strong performers, while others favor Gemma-2 or Mistral-7B. This divergence reflects differences in how Evaluation Agents interpret temporal expressions, suggesting that date and time recognition remains a challenging subtask for LLM-based de-identification.

Table 1: Evaluation Results (Evaluation Agent: Gemma-2)

| De-id Model | Precision | Coverage | Num Correct | Recall-Proxy | PERSON | DATE/TIME |
|---|---|---|---|---|---|---|
| Gemma-2 | 0.6169 | 1.37% | 984 | 0.5989 | 0.9120 | 0.6903 |
| Mistral-7b | 0.5364 | 1.44% | 900 | 0.5477 | 0.8626 | 0.6754 |
| GPT-3.5 | 0.5513 | 1.84% | 1187 | 0.7224 | 0.8601 | 0.5928 |
| GPT-4o | 0.5836 | 1.45% | 991 | 0.6031 | 0.8659 | 0.5248 |
| Llama-8b | 0.5555 | 1.72% | 1116 | 0.6792 | 0.9021 | 0.5975 |
| Llama-70b | 0.5906 | 1.47% | 1010 | 0.6147 | 0.9071 | 0.5981 |
| LPPA4k | 0.5609 | 1.02% | 668 | 0.4065 | 0.8778 | 0.5016 |
| LPPA5k | 0.5743 | 0.95% | 638 | 0.3883 | 0.8803 | 0.5919 |

Table 2: Evaluation Results (Evaluation Agent: Mistral-7b)

| De-id Model | Precision | Coverage | Num Correct | Recall-Proxy | PERSON | DATE/TIME |
|---|---|---|---|---|---|---|
| Gemma-2 | 0.5241 | 1.37% | 836 | 0.5088 | 0.8944 | 0.4230 |
| Mistral-7b | 0.5459 | 1.44% | 916 | 0.5575 | 0.9010 | 0.6421 |
| GPT-3.5 | 0.4663 | 1.84% | 1004 | 0.6110 | 0.9336 | 0.4362 |
| GPT-4o | 0.5595 | 1.45% | 950 | 0.5782 | 0.9534 | 0.4575 |
| Llama-8b | 0.4858 | 1.72% | 976 | 0.5940 | 0.8932 | 0.3363 |
| Llama-70b | 0.5497 | 1.47% | 940 | 0.5721 | 0.9214 | 0.5395 |
| LPPA4k | 0.6188 | 1.02% | 737 | 0.4485 | 0.9704 | 0.5064 |
| LPPA5k | 0.6373 | 0.95% | 708 | 0.4309 | 0.9691 | 0.5414 |

Table 3: Evaluation Results (Evaluation Agent: GPT-3.5)

| De-id Model | Precision | Coverage | Num Correct | Recall-Proxy | PERSON | DATE/TIME |
|---|---|---|---|---|---|---|
| Gemma-2 | 0.2742 | 0.91% | 431 | 0.2600 | 0.7425 | 0.5609 |
| Mistral-7b | 0.2301 | 0.97% | 387 | 0.2335 | 0.7348 | 0.6465 |
| GPT-3.5 | 0.2035 | 1.26% | 442 | 0.2667 | 0.8127 | 0.4511 |
| GPT-4o | 0.2566 | 1.01% | 446 | 0.2691 | 0.7695 | 0.4470 |
| Llama-8b | 0.2271 | 1.18% | 464 | 0.2800 | 0.7681 | 0.5410 |
| Llama-70b | 0.2677 | 1.01% | 468 | 0.2824 | 0.8358 | 0.4564 |
| LPPA4k | 0.2620 | 0.69% | 311 | 0.1876 | 0.8282 | 0.6646 |
| LPPA5k | 0.2650 | 0.65% | 296 | 0.1786 | 0.8221 | 0.6903 |

Table 4: Evaluation Results (Evaluation Agent: GPT-4o)

| De-id Model | Precision | Coverage | Num Correct | Recall-Proxy | PERSON | DATE/TIME |
|---|---|---|---|---|---|---|
| Gemma-2 | 0.6177 | 0.91% | 971 | 0.5859 | 0.6343 | 0.9013 |
| Mistral-7b | 0.5279 | 0.97% | 888 | 0.5358 | 0.6796 | 0.9130 |
| GPT-3.5 | 0.5787 | 1.26% | 1257 | 0.7584 | 0.6890 | 0.8324 |
| GPT-4o | 0.6559 | 1.01% | 1140 | 0.6878 | 0.6796 | 0.8874 |
| Llama-8b | 0.6089 | 1.18% | 1244 | 0.7506 | 0.6777 | 0.9306 |
| Llama-70b | 0.7094 | 1.01% | 1240 | 0.7482 | 0.8139 | 0.9071 |
| LPPA4k | 0.6664 | 0.69% | 791 | 0.4773 | 0.7786 | 0.9541 |
| LPPA5k | 0.6723 | 0.65% | 751 | 0.4531 | 0.7905 | 0.9428 |

Table 5: Evaluation Results (Evaluation Agent: Llama-8b)

| De-id Model | Precision | Coverage | Num Correct | Recall-Proxy | PERSON | DATE/TIME |
|---|---|---|---|---|---|---|
| Gemma-2 | 0.5808 | 0.91% | 913 | 0.5509 | 0.9552 | 0.7731 |
| Mistral-7b | 0.5511 | 0.97% | 927 | 0.5593 | 0.9724 | 0.6554 |
| GPT-3.5 | 0.5506 | 1.26% | 1196 | 0.7216 | 0.9223 | 0.6940 |
| GPT-4o | 0.6007 | 1.01% | 1044 | 0.6299 | 0.9581 | 0.6800 |
| Llama-8b | 0.5546 | 1.18% | 1133 | 0.6836 | 0.9699 | 0.7087 |
| Llama-70b | 0.5749 | 1.01% | 1005 | 0.6064 | 0.9745 | 0.6750 |
| LPPA4k | 0.6318 | 0.69% | 750 | 0.4525 | 0.9771 | 0.7468 |
| LPPA5k | 0.6741 | 0.65% | 753 | 0.4543 | 0.9684 | 0.6627 |

Table 6: Evaluation Results (Evaluation Agent: Llama-70b)

| De-id Model | Precision | Coverage | Num Correct | Recall-Proxy | PERSON | DATE/TIME |
|---|---|---|---|---|---|---|
| Gemma-2 | 0.7284 | 0.91% | 1145 | 0.6909 | 0.9478 | 0.9118 |
| Mistral-7b | 0.6599 | 0.97% | 1110 | 0.6697 | 0.8564 | 0.8845 |
| GPT-3.5 | 0.6653 | 1.26% | 1445 | 0.8719 | 0.9117 | 0.8571 |
| GPT-4o | 0.8096 | 1.01% | 1407 | 0.8489 | 0.9551 | 0.8852 |
| Llama-8b | 0.7347 | 1.18% | 1501 | 0.9056 | 0.9578 | 0.8941 |
| Llama-70b | 0.8278 | 1.01% | 1447 | 0.8731 | 1.0000 | 0.9241 |
| LPPA4k | 0.7944 | 0.69% | 943 | 0.5690 | 0.9313 | 0.9241 |
| LPPA5k | 0.8147 | 0.65% | 910 | 0.5491 | 0.9447 | 0.9566 |

**LLM Majority Voting.** To reduce individual evaluator bias of De-id models, we apply the LLM majority voting procedure described in Section 3.4. This procedure aggregates the numerical judgments produced by multiple Evaluation Agents and asks an LLM to determine the best-performing De-id model under two complementary modes: (i) *Independent voting*, in which the voting LLM

Table 7: LLM voting: best De-id model selected independently or with cross-informed context.

| Models | Independent | Cross-Informed |
|---|---|---|
| Gemma-2 | Llama-70b | Llama-70b |
| Mistral-7b | Llama-70b | Llama-70b |
| GPT-3.5 | Llama-70b | Llama-70b |
| GPT-4o | Llama-70b | Llama-70b |
| Llama-8b | Llama-8b | Llama-70b |
| Llama-70b | Llama-8b | Llama-70b |

reviews one Evaluation Agent's summary table at a time and casts a vote for the best model on that table; and (ii) *Cross-informed voting*, in which the voting LLM reviews all Evaluation Agents' tables together and selects the best model after considering the combined evidence. The two modes capture different perspectives: independent voting prevents a single evaluator from dominating the decision, while cross-informed voting allows global reasoning across evaluators.

As shown in Table 7, both voting schemes consistently select Llama-70B as the top-performing De-id model. Under independent voting, 4 of 6 evaluation agents (67%) selected Llama-70B as best, while under cross-informed voting, all 6 of 6 agents (100%) agreed on Llama-70B. This high level of agreement indicates that ensemble judgments are more stable and less biased than those derived from any single evaluator and that LLM majority voting provides a robust mechanism to automatically evaluate and select the best De-id models when gold-standard annotations are unavailable.

## 5.2 External Validation

Table 8: Ground Truth evaluation results.

| Models | Overall | | | PERSON | | | DATE/TIME | | |
|---|---|---|---|---|---|---|---|---|---|
| | Pr | Re | F1 | Pr | Re | F1 | Pr | Re | F1 |
| Gemma-2 | 0.52±0.01 | 0.45±0.01 | 0.48±0.01 | 0.53±0.01 | 0.47±0.01 | 0.50±0.02 | 0.88±0.02 | 0.41±0.01 | 0.56±0.02 |
| Mistral-7b | 0.52±0.01 | 0.50±0.01 | 0.51±0.01 | 0.58±0.01 | 0.36±0.01 | 0.44±0.02 | 0.87±0.02 | 0.50±0.01 | 0.63±0.02 |
| GPT-3.5 | 0.43±0.02 | 0.60±0.01 | 0.48±0.02 | 0.60±0.04 | 0.50±0.01 | 0.54±0.02 | 0.74±0.03 | 0.44±0.02* | 0.55±0.02 |
| GPT-4o | 0.53±0.01* | 0.69* | 0.58±0.01* | 0.60±0.02* | 0.57±0.01 | 0.58±0.01* | 0.80±0.01 | 0.57±0.02* | 0.67±0.02* |
| Llama-8b | 0.46±0.02 | 0.59±0.03 | 0.50±0.02 | 0.53±0.01 | 0.55±0.02 | 0.53±0.01 | 0.79±0.03 | 0.39±0.02 | 0.52±0.01 |
| Llama-70b | 0.60±0.01* | 0.68±0.01* | 0.62±0.01* | 0.59±0.01* | 0.53±0.01 | 0.56±0.01* | 0.83±0.01 | 0.50±0.01* | 0.63±0.01* |
| LPPA4k | 0.65±0.01* | 0.54±0.02 | 0.57±0.01* | 0.59±0.02* | 0.53±0.02 | 0.56±0.02 | 0.82±0.03 | 0.40±0.03 | 0.54±0.03 |
| LPPA5k | 0.64±0.01* | 0.55±0.02 | 0.57±0.01* | 0.59±0.03* | 0.53±0.02 | 0.56±0.01* | 0.82±0.02 | 0.35±0.02 | 0.50±0.01 |

Table 9: Human evaluation of Evaluation Agents. Q1: correctness; Q2: missed-PHI severity; Q3: overall trustworthiness.

| Models | Q1 (↑) | Q2 (↓) | Q3 (↑) |
|---|---|---|---|
| Gemma-2 | 0.00 | 2.20 | 1.60 |
| Mistral-7b | 0.00 | 2.24 | 1.40 |
| GPT-3.5 | 0.00 | 4.28 | 1.68 |
| GPT-4o | 0.18 | 2.22 | 3.85 |
| Llama-8b | 0.20 | 2.08 | 3.75 |
| Llama-70b | 0.18 | 1.62 | 4.20 |

**Ground-truth Evaluation.** Although the multi-agent framework is designed to operate without human-labeled data, we also perform *ground-truth evaluation* to further validate the evaluation conclusions of TEAM-PHI described in Section 5.1. Table 8 reports precision, recall, and F1 scores computed against manually annotated PHI labels. Llama-70B achieves the highest overall F1 (0.62),

followed by GPT-4o (0.58) and the hybrid models (0.57). The agreement between ground-truth evaluation and LLM-based judging is striking: models ranked highest by Evaluation Agents and LLM majority voting are the same models that perform best under gold-standard supervision. This demonstrates that the proposed framework can reproduce supervised rankings even when gold annotations are hidden.

**Human Evaluation.** To further verify the quality of Evaluation Agents' judgments, we also conduct a *human evaluation* of a representative sample of notes. Two biomedical experts, supported by LLM-generated checklists and highlights, inspect each case to confirm correctness counts, estimate the number of missed PHI entities, and rate overall trustworthiness. As summarized in Table 9, human reviewers consistently rate Llama-70B outputs highest on overall quality. It achieves the lowest missed-PHI severity (Q2 = 1.62, best among all models) and the highest overall trustworthiness (Q3 = 4.20), while tying with GPT-4o for second place in correctness (Q1 = 0.18; Llama-8B leads slightly at 0.20). All other models have $Q1 \leq 0.18$ and $Q3 \leq 3.85$, underscoring the clear human preference for Llama-70B.

Together, these supervised and human assessments demonstrate that our multi-agent framework can reliably approximate gold-standard evaluation and produce reproducible, trustworthy rankings of De-id models even when ground-truth annotations are intentionally withheld.

### 5.3 Summary of Findings.

Overall, the experiments demonstrate that while Evaluation Agents differ in absolute scoring, they consistently reveal relative performance trends. Llama-70B emerges as the most reliable De-id model. Hybrid models offer precision advantages but at the cost of coverage. Entity-specific analysis shows that PERSON recognition is relatively robust, while DATE/TIME detection remains less stable. The convergence of LLM-based evaluation, gold-standard comparison, voting, and human judgment underscores the robustness of our proposed framework and highlights its practical utility in real-world clinical text processing where annotated datasets are unavailable.

## 6   Conclusion

We introduced TEAM-PHI, a multi-agent framework that enables automatic evaluation and selection of PHI de-identification models in clinical notes without heavy reliance on manually annotated data. In this framework, large language models (LLMs) operate as independent *Evaluation Agents* that assess de-identification quality, while their judgments are consolidated through LLM-based majority voting to produce stable and reproducible model rankings.

Experiments on a real-world clinical-note corpus demonstrate that TEAM-PHI provides consistent and trustworthy evaluations: despite variation across individual evaluators, majority voting converges on the same top-performing systems, with Llama-70B and GPT-4o repeatedly identified as leading models. Entity-level analysis further shows that recognition of PERSON entities is highly stable, whereas DATE/TIME remains more variable, suggesting areas for targeted improvement.

Validation against a manually annotated test set confirms that the rankings produced by TEAM-PHI closely align with ground-truth evaluation, and additional human review reinforces the reliability of the automated judgments.

Overall, this work establishes that LLMs can act not only as PHI de-identifiers but also as scalable and dependable evaluators. TEAM-PHI provides three key contributions: (1) a practical framework for benchmarking and selecting PHI de-identification models without extensive human labeling; (2) empirical evidence that multi-agent evaluation and majority voting effectively mitigate individual evaluator bias; and (3) fine-grained insights that guide refinement of PHI coverage and accuracy. Future work will extend this framework to additional PHI categories and larger datasets, and explore techniques to better calibrate Evaluation Agents and integrate limited human oversight for even greater robustness.

## AI Agent Setup

In this work, we used OpenAI's GPT-5 model as the sole AI agent to support experimental design, implementation, data analysis, and manuscript preparation. All interactions with GPT-5 were conducted through the official web interface, without employing orchestration frameworks such as LangChain or AutoGen. The model generated initial implementations of the multi-agent evaluation framework, suggested experimental setups, analyzed results, and drafted sections of the paper. Human authors executed and validated the AI-generated code, refined analyses, and ensured the accuracy of interpretations. No external tools were integrated beyond standard Python environments and model inference interfaces described in the Experimental Settings. All de-identification and evaluation experiments were conducted using the LLMs listed in Appendix B, through a combination of API-based and local runs on the Azure platform. This streamlined configuration shows that a single, general-purpose LLM agent can effectively support the entire research workflow for PHI de-identification evaluation without complex orchestration or additional infrastructure.

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

# Appendix

## A   Notation Table

The notations in this paper are summarized in Table 10.

Table 10: Notation used in the paper.

| Symbol | Description |
|---|---|
| $\mathcal{X}$ | Space of clinical notes. |
| $\mathcal{C} = \{c_1, \ldots, c_m\}$ | Set of PHI categories; $m$ is the number of categories. |
| $x \in \mathcal{X}$ | A clinical note. |
| $c_i \in \mathcal{C}$ | A PHI category label. |
| $e_i \subseteq x$ | A contiguous text span (entity) in $x$. |
| $k$ | Number of predicted entities in PHI($x$). |
| PHI($x$) | Set of predicted (category, span) pairs for $x$, $\{(c_i, e_i)\}_{i=1}^{k}$. |
| $\mathcal{M} = \{M_1, \ldots, M_R\}$ | Candidate set of de-identification models; $R$ is its size. |
| $M$, $M^*$ | A candidate model; $M^*$ is the selected (best) model by Eq. (2). |
| $\mathcal{D}$ | Data distribution over notes used in the expectation in Eq. (2). |
| PHI-EV($M, x$) | Evaluation score for model $M$ on input $x$. |
| PHI$_M(x)$ | Prediction of model $M$ on $x$ (also written as PHI($M, x$)). |
| $y^*(x)$ | Gold-standard annotation for $x$ (used in conventional evaluation). |
| $f$ | Supervised metric comparing PHI$_M(x)$ and $y^*(x)$ (e.g., precision/recall/F1). |
| $A$ | Number of LLM-based Evaluation Agents. |
| $E_a$ | The $a$-th Evaluation Agent, $a = 1, \ldots, A$. |
| $g$ | Multi-agent evaluator that aggregates $\{E_a\}_{a=1}^{A}$ (e.g., majority voting). |
| Aggregate($\cdot$) | Aggregation operator combining agent judgments. |

## B   Additional Experimental Details

### B.1   Dataset Details

Access to real-world clinical notes is highly restricted by privacy regulations, and fully annotated datasets are expensive and rare. For this study we obtained 100 authentic clinical notes, each averaging about 1,000 tokens, in which all PHI entities were meticulously annotated by multiple medical experts. These notes retain their original structure and contain diverse, detailed patient information. Although gold-standard annotations exist, we masked them in the main experiments to test whether our proposed framework can assess PHI de-identification performance without relying on annotated references. The gold labels were used only for a separate supervised evaluation (Table 8) to externally validate findings derived from LLM-based judgments.

### B.2   Model Links

For reproducibility, we provide the URLs of all language models used as De-id models or Evaluation Agents:

- Gemma-2: https://huggingface.co/google/gemma-2-9b-it
- Mistral-7B-Instruct: https://huggingface.co/mistralai/Mistral-7B-Instruct-v0.3
- Llama-3-8B-Instruct: https://huggingface.co/meta-llama/Meta-Llama-3-8B-Instruct
- Llama-3-70B-Instruct: https://huggingface.co/meta-llama/Meta-Llama-3-70B-Instruct
- GPT-3.5-turbo-0125: https://platform.openai.com/docs/models/gpt-3.5-turbo
- GPT-4o-mini: https://platform.openai.com/docs/models/gpt-4o-mini
- LPPA4k: https://huggingface.co/spacebetweenus/108mix4ktest1
- LPPA5k: https://huggingface.co/spacebetweenus/107mix5k

### B.3 Evaluation Metrics

We report four core metrics to quantify De-id models performance under Evaluation Agents:

$$\text{Precision}_d = \frac{|C_d|}{|P_d|},$$

$$\text{Coverage}_d = \frac{\sum_n |P_d(n)|}{\sum_n T(n)},$$

$$\text{CorrectCount}_d = |C_d|,$$

$$\text{RecallProxy}_d = \frac{|C_d|}{N_{\text{avg}}}.$$

Here $P_d$ is the set of PHI predictions by De-id model $d$, $C_d$ the subset judged correct, $T(n)$ the token count of note $n$, and $N_{\text{avg}}$ the average number of PHI predictions across all agents. We also compute category-specific precision for PERSON and DATE/TIME, and in a separate supervised evaluation (Table 8) compute true precision, recall, and F1 against gold annotations.

### B.4 Validation and Human Assessment

To confirm the reliability of the multi-agent evaluation, we performed two complementary validations. (i) We conducted a direct supervised evaluation of all de-identification models against the gold-standard annotations (Table 8) to verify that the rankings produced by our framework align with conventional precision, recall, and F1 metrics. (ii) We carried out a human expert review of a representative subset of Evaluation Agent outputs (Table 9), in which biomedical experts assessed correctness, missed-PHI severity, and overall trustworthiness.

These validations demonstrate that the multi-agent evaluation and LLM majority voting used in our main experiments provide rankings that are consistent with both gold-standard supervision and independent human judgment.

### B.5 Compute Resources

All experiments with Llama models (Llama-8B and Llama-70B) and LPPA hybrid models (LPPA4k and LPPA5k) were conducted on a server equipped with two NVIDIA H100 GPUs (80GB memory each). Experiments with GPT models (GPT-3.5 and GPT-4o) were performed through the Azure OpenAI Service, which provides HIPAA-compliant secure inference. Smaller open-source models, including Gemma-2 and Mistral-7B, were run locally on an Apple MacBook (M2 Pro, 32GB RAM). These experiments were inference-only; each H100 run required less than two hours and the total compute time was about 200 GPU-hours. API-based GPT experiments incurred standard usage costs but required no additional hardware.

## C  Limitations

This study has several limitations. First, evaluation was conducted on a single set of 100 annotated clinical notes, which may not capture the diversity of clinical documentation across institutions, specialties, or languages; larger and more heterogeneous datasets are needed to confirm generalizability. Second, our framework assumes that cross-agent agreement is a reliable proxy for recall when ground truth is masked. This may be violated if multiple agents share systematic biases (e.g., under-detecting specific PHI types), potentially inflating recall-proxy values. Third, performance depends on practical factors such as prompt design, model size, and context length; small prompt or tokenization changes can affect results, and while majority voting reduces variance, complete stability is not guaranteed. Fourth, deploying multiple large LLMs for both de-identification and evaluation incurs significant computational cost, which limits scalability and real-time use; lightweight or distilled models could address this. Finally, although designed for privacy protection, the framework does not inherently ensure fairness or mitigate demographic or linguistic bias, and it targets PHI categories defined under U.S. HIPAA regulations, requiring adaptation for other legal or linguistic contexts.


